# Early Gastrointestinal Neuropathy Assessed by Wireless Motility Capsules in Adolescents with Type 1 Diabetes

**DOI:** 10.3390/jcm12051925

**Published:** 2023-02-28

**Authors:** Vinni Faber Rasmussen, Mathilde Thrysøe, Páll Karlsson, Esben Thyssen Vestergaard, Kurt Kristensen, Ann-Margrethe Rønholt Christensen, Jens Randel Nyengaard, Astrid Juhl Terkelsen, Christina Brock, Klaus Krogh

**Affiliations:** 1Danish Pain Research Center, Department of Clinical Medicine, Aarhus University, 8200 Aarhus, Denmark; 2Department of Pediatrics and Adolescent Medicine, Randers Regional Hospital, 8930 Randers, Denmark; 3Steno Diabetes Center Aarhus, Aarhus University Hospital, 8200 Aarhus, Denmark; 4Core Centre for Molecular Morphology, Section for Stereology and Microscopy, Department of Clinical Medicine, Aarhus University, 8200 Aarhus, Denmark; 5Department of Pediatrics and Adolescent Medicine, Aarhus University Hospital, 8200 Aarhus, Denmark; 6Department of Pediatrics and Adolescent Medicine, Aalborg University Hospital, 9000 Aalborg, Denmark; 7Steno Diabetes Center North Denmark, 9000 Aalborg, Denmark; 8Department of Pathology, Aarhus University Hospital, 8200 Aarhus, Denmark; 9Department of Neurology, Aarhus University Hospital, 8200 Aarhus, Denmark; 10Department of Gastroenterology, Aalborg University Hospital, 9000 Aalborg, Denmark; 11Department of Hepatology and Gastroenterology, Aarhus University Hospital, 8200 Aarhus, Denmark

**Keywords:** adolescent, type 1 diabetes, wireless motility capsule, gastrointestinal symptoms, autonomic neuropathy

## Abstract

Background: To assess the prevalence of objective signs of gastrointestinal (GI) autonomic neuropathy (AN) in adolescents with type 1 diabetes (T1D). In addition, to investigate associations between objective GI findings and self-reported symptoms or other findings of AN. Methods: Fifty adolescents with T1D and 20 healthy adolescents were examined with a wireless motility capsule to assess the total and regional GI transit times and motility index. GI symptoms were evaluated with the GI Symptom Rating Scale questionnaire. AN was evaluated with cardiovascular and quantitative sudomotor axon reflex tests. Results: There was no difference in GI transit times in adolescents with T1D and healthy controls. Adolescents with T1D had a higher colonic motility index and peak pressure than the controls, and GI symptoms were associated with low gastric and colonic motility index (all *p* < 0.05). Abnormal gastric motility was associated with the duration of T1D, while a low colonic motility index was inversely associated with “time in target range” for blood glucose (all *p* < 0.01). No associations were found between signs of GI neuropathy and other measures of AN. Conclusions: Objective signs of GI neuropathy are common in adolescents with T1D and it seems to require early interventions in patients at high risk of developing GI neuropathy.

## 1. Introduction

Gastrointestinal (GI) symptoms are prevalent in individuals with type 1 diabetes (T1D) and negatively impair their quality of life [1,2]. Common symptoms include abdominal pain, dyspepsia, reflux, poor appetite, postprandial fullness, swallowing difficulties, nausea, vomiting, diarrhea, chronic constipation, and fecal incontinence [1,3,4]. Usually, severe GI symptoms appear after a minimum of 10 years of having the disease, making early intervention crucial for preventing further progression and worsening of symptoms. Although GI symptoms are also common in healthy adolescents, they are not more frequent in adolescents with T1D [1,4]. Autonomic neuropathy and hyperglycemia are the most widely recognized contributing factors to GI symptoms in T1D [5,6]. Therefore, the identification of subclinical signs of GI neuropathy could be of clinical importance for the individual patient. By identifying risk factors for subclinical GI neuropathy, targeted investigations can be performed in high-risk adolescents without exposing the entire population of young individuals with T1D to time-consuming clinical tests. In adolescents with T1D, GI symptoms are associated with increased HbA1c, duration of diabetes, insulin requirement, body mass index (BMI), poor socioeconomic status, daily cigarette smoking, and an irregular meal pattern [1,4]. The relationship between these factors and early objective signs of GI neuropathy is yet to be determined. Potential early interventions such as improved blood glucose control and gastroparesis treatment may have positive outcomes [7].

Up to 75% of adolescents with T1D may show signs of autonomic dysfunction, depending on the diagnostic methods and definitions used [8,9,10]. However, it is unclear how accurately tests for neuropathy in other organs can predict neuropathy in the GI tract, which also includes damage to the enteric nervous system. Because GI neuropathy in T1D potentially affects all regions of the GI tract, it is essential to perform a panenteric assessment of GI function [11,12]. While several methods qualify for this [13], few have been applied to adolescents with T1D. Perano et al. performed a (13)C-octanoate breath test to evaluate the relationship between gastric emptying time and postprandial glycemia, but not for evaluating the signs of neuropathy [14].

Capsule-based methods are generally considered the most appropriate for studying GI neuropathy in diabetes and other severe motility disorders [11,13]. They provide a minimally invasive assessment of pan-enteric motility, including total and regional GI transit times and contractility patterns [11,13]. The wireless motility capsule (WMC) (Smartpill Monitoring System, Medtronic, MN, USA) has been tested in people with diabetes and is commercially available. It is easy to use and has robust normative data for healthy adults [11,15]. This argues for the use of WMC as a suitable method for assessing GI neuropathy in both research and clinical settings for adolescents with T1D.

The aims of the present study were to (1) investigate the prevalence of signs of GI neuropathy in a selected group of adolescents with T1D and to compare them to healthy age-matched participants; (2) investigate the association between GI transit times and motility index to self-reported GI symptoms, as well as findings on tests for cardiovascular and sudomotor function; and (3) identify potential risk factors for early GI autonomic neuropathy in adolescents with T1D.

## 2. Materials and Methods

### 2.1. Study Population

The study was a part of the T1DANES study. Adolescents aged 15 to less than 19 years old with T1D and a history of diabetes for at least five years were recruited from outpatient clinics at Danish hospitals in Randers, Aarhus, and Aalborg, as well as the Steno Diabetes Centre Aarhus and North Denmark, from August 2020 to December 2021. Exclusion criteria were participants who were prescribed medication or who had other diseases that could affect the central or peripheral nervous system. Additionally, a negative COVID-19 test result within 72 h before the test day was required. Participants with well-treated autoimmune disorders such as thyroid disease or celiac disease, or complications to diabetes such as microalbuminuria were accepted and included in the study. Healthy age-matched controls were recruited through notices at boarding and secondary schools.

Information about age, gender, diabetes duration, total daily insulin dose, basal insulin dose, time-in-range, glucose-monitoring system, HbA1c values over the last five years, events of severe hypoglycemia and ketoacidosis during the last year, and the last test results of retinopathy and nephropathy (urine albumin/creatinine ratio) was obtained from the patients’ clinical electronic records covering all outpatient visits.

Informed oral and written consent was obtained from each participant and the accompanying parents. All procedures in the study protocol were approved by the Danish Ethics Committee (Project ID M-2019-211-19) and Legal Office, Central Denmark Region (1-16-02-42-21). Data were safely stored in REDCap, a secure web application for online surveys and databases.

### 2.2. Clinical and Biochemical Data Collection

All of the available clinical and biochemical data were extracted from the electronic hospital records of the adolescents, and any missing data were collected on the test day. A blood sample was taken from each participant for later analysis. The blood from healthy controls was analyzed for HbA1c and lipid profile. Participants arrived at the research facility at Aarhus University Hospital in the morning, after fasting for at least 6 h for food and nutrient-containing liquid (milk) and 2 h for water. Caffeine (coffee and cola) and alcohol were not allowed for 12 h before the 08:00 a.m. meeting time. The weight and height of each participant were measured, and the BMI (kg/m^2^) was calculated. Hip and waist measurements were taken using a measuring tape, and blood pressure and heart rate were recorded using an automatic blood pressure monitor. The puberty stage was assessed by showing the participants pictures of different Tanner stages and asking them to point to the relevant stage. The participants also self-reported their activity levels, alcohol consumption, and smoking status.

### 2.3. Questionnaires

The GI Symptom Rating Scale (GSRS), a 15-item instrument with questions into five symptom domains, reflux, abdominal pain, indigestion, diarrhea, and constipation, was filled out online, at home, before the study day [16].

### 2.4. Wireless Motility Capsule

WMC was used to describe gastric motility and regional GI transit times. Prior to the test day, the adolescents with T1D were informed to take their regular basal insulin dose in the morning and bring their blood glucose levels within the target range (4–8 mmol/L).

On the test day, the participants consumed a standard meal (SmartBar), and they were allowed to drink a maximum of 200 mL of water to swallow the capsule. When ingested, the capsule travelled through the GI tract and transmitted information about temperature, pH, and pressure to a receiver worn on the abdomen. The patients carried the receiver until the capsule was expelled, normally within one to five days. The imported data provided valid information about the segmental transit times and motility index [13,15,17,18]. Data from the adolescents with T1D were compared to previously published normal ranges for transit times in adults (upper limits: gastric emptying 5 h, small bowel transit 8 h, colonic transit 50.5 h [19]) and to the motility index in adults [11]. In addition, data from the adolescents with T1D were compared to data from healthy adolescents in the present study.

### 2.5. Autonomic Tests

The evaluation of cardiovagal function was carried out using the following cardiovascular reflex tests (CARTs) [20]: (1) deep breathing test, which measures the delta heart rate and the difference in heart rate between expiration and inspiration; (2) the Valsalva maneuver (VM) ratio, obtained from forcefully exhaling with expiratory pressure of 40 mmHg for 15 s in a 20-degree tilt position; and (3) the response to standing, measured using the 30:15 ratio. The autonomic tests were performed in a standardized manner using a Task Force Monitor^®^ (CNSystems Medizintechnik AG, Graz, Austria), obtained from a three-channel electrocardiogram. Real-time respiratory pressure and volume were measured by blowing into a mouthpiece connected to a digital transducer.

A quantitative sudomotor reflex test (QSART) [21] was performed on the right side of the body at four locations: the forearm, proximal leg, distal leg, and on the foot, under a heat lamp so as to maintain a constant temperature. The nerves were stimulated with acetylcholine, and the test was conducted using WR TestWorks Q-Sweat Quantitative Sweat Measurement System (WR Medical Electronics Co., Maplewood, MN, USA).

The data from the autonomic testing will be presented in a separate publication.

### 2.6. Statistical Analysis

All statistical analyses were conducted using the software program R (R Core Team (2022), Vienna, Austria). The normality of the variables in Table 1 were tested using the Shapiro–Wilk test and QQ plots. Descriptive data are presented as the mean (SD) for normally distributed continuous variables, median (range) for non-normally distributed continuous variables, and number (%) for categorical variables. The groups were compared using Student’s *t*-test for continuous variables with normal distribution, Wilcoxon rank-sum test for non-parametric continuous variables, and Fisher’s exact test for categorical variables. *p*-Values of less than 0.05 were considered statistically significant. The abnormality of a diagnostic test was defined as below the 5th (or above the 95th) percentile of the data obtained from the control subjects. Linear regression (lm() function in R) was applied to analyze the associations, and ROC analysis with area under the curve (AUC) was used to evaluate the usefulness of the tests as screening methods, with AUC values of 0.5 indicating no discrimination, 0.5–0.7 poor discrimination, 0.7–0.8 acceptable discrimination, 0.8–0.9 excellent discrimination, and greater than 0.9 outstanding discrimination.

## 3. Results

Fifty-five adolescents with T1D and twenty-one healthy adolescents were included in this study. A flowchart of the selection process is shown in Figure 1. One healthy adolescent was excluded due to having a whole gut transit time exceeding five days. The median duration of T1D among 55 adolescents was 8.5 (range: 5–17 years), and their HbA1c was 61 mmol/mol (range: 41–93 mmol/mol). See Table 1 for additional details on the characteristics of the two groups.

Overall, there was no difference in the total or regional GI transit times between healthy adolescents and those with T1D (Table 2). Four adolescents with T1D (8%) had one or more segmental transit times that exceeded the upper 95% percentile range of the data collected from our healthy controls in the study. This included two with a prolonged gastric emptying time, one with a prolonged small intestinal transit time, and one with a prolonged colonic transit time.

Peak pressure amplitude (*p* = 0.010) and motility index (*p* = 0.022) in the colon were higher in the adolescents with T1D than in the controls (as shown in Table 2). Six adolescents had a low motility index of the small intestine (below the 5th percentile of healthy adolescents), and four adolescents had a low motility index in the colon.

When comparing healthy adolescents to those with T1D, there was no difference in total GSRS score (median (range) 1.35 (1–2.87) vs. 1.24 (1–2.97), *p* = 0.09).

A low colonic motility index was associated with both severe diarrhea (*p* = 0.042) and indigestion (*p* = 0.038), as assessed by GSRS. Additionally, a higher total GSRS score was associated with both a low gastric motility index (*p* = 0.047) and low colon motility index (*p* = 0.033). The ROC analysis of the total GSRS score as a screening tool for predicting abnormal GI motility showed an AUC of 0.67, which was considered “acceptable”.

No associations were found between the GI parameters (transit times, motility index, and GSRS scores) and autonomic test results (CARTs and QSART) (data not shown, all *p* > 0.05).

When evaluating the clinical factors, we found that diabetes duration and “time in range” for blood glucose were risk factors. A longer gastric emptying time (*p* = 0.004) and higher gastric motility index (*p* = 0.009) were associated with the time since diagnosis of T1D. The colon motility index was inversely associated with “time in range” for blood glucose (*p* = 0.003).

Of the four included adolescents with T1D who had microvascular complications (as listed in Table 1), only one had an abnormal finding (prolonged colonic transit time) on the WMC. The only included adolescents with T1D and well-treated celiac disease had a prolonged gastric emptying time.

## 4. Discussion

GI symptoms are a major concern for many individuals with T1D [22,23]. The present study is the first to show that objective signs of gastroenteric neuropathy are prevalent in adolescents with T1D and a history of diabetes for at least five years. Typically, major changes in GI motility are indicated by abnormal gastric emptying or altered small intestinal or colonic transit times, but this was not observed in this study. Instead, discrete changes in contractility patterns were found more in adolescents with T1D, suggesting early stages of GI neuropathy. This is supported by the association we found between objective signs of GI neuropathy, the length of time since the onset of T1D, and poor metabolic control.

The neuronal control of GI motility is complex. The frequency of contractions is controlled by the interstitial cells of the Cajal, also named GI pacemaker cells. Interneurons within the enteric nervous system connect the Cajal cells with the smooth muscle cells of the GI tract. Most excitatory interneurons are cholinergic. The sympathetic nervous system inhibits GI motility in non-sphincteric regions, and the parasympathetic nervous system enhances it via the vagal nerves or sacral segments of the spinal cord. This difference in innervation is probably important for the understanding of our results. In general, GI motility indices were higher in our group of adolescents with T1D compared with the healthy controls. This could indicate a loss of inhibitory control from the sympathetic nervous system. Conversely, a lower “time in range” for blood glucose was associated with a higher motility index in the colon, potentially due to elevated blood glucose on the day of investigation or a progression of GI neuropathy affecting parasympathetic nerves. Progression to neuropathy of cholinergic nerves seems likely and may be of clinical relevance, as diarrhea and indigestion were associated with reduced motility in the colon. This would either imply reduced activity of the Cajal cells or neuropathy of cholinergic nerves. In addition to neuropathy, factors such as morphological changes in the gut wall and disturbed blood glucose control likely contribute to symptoms [23].

### 4.1. Methods for Assessment of Enteric Neuropathy

Our findings, along with previous studies, suggest that the GSRS questionnaire could serve as a screening tool for gastroenteric neuropathy. Although several objective methods exist for evaluating gastroenteric neuropathy, most focus on gastric emptying [13]. However, as diabetes can impact the entire GI tract, a pan-enteric evaluation is necessary [24]. Our study supports this by showing that different segments of the GI tract can be affected and abnormalities can vary from patient to patient. One important limitation of WMC as a diagnostic method for autonomic neuropathy is that it only provides information on intestinal pressure rather than detecting peristaltic waves and responses to nerve stimuli [13]. However, WMC has been proven to be highly sensitive in symptomatic pediatric patients [25].

The 11C-donepezil PET/CT scan visualizes cholinergic innervation of the GI tract and has shown reduced cholinergic activity in the gut of patients with T1D [13]. Unfortunately, this test is only available at a few centers and is not suitable for children as a result of radiation exposure.

Another option for adults is high-resolution manometry, which assesses antro-duodenal or colonic motor function. This test involves the placement of a catheter through colonoscopy, which can disrupt normal physiology. The Motilis 3D-Transit system is a superior alternative, as it provides exact information about the location of the capsule in the GI tract though an ambulatory electromagnetic wireless capsule system. Although the Motilis 3D-Transit system has been trialed in children [26], it is not yet clinically available and is only offered at specialist centers [27].

Vagal nerve function can also be assessed by measuring the pancreatic polypeptide and ghrelin in response to Sham feeding. Previous studies have shown impaired pancreatic polypeptide response in diabetic gastroparesis, indicating dysfunction of the vagal nerve [28].

### 4.2. Autonomic Evaluation

In clinical practice, tests of cardiovascular reflexes are sometimes taken as a surrogate for tests of enteric neuropathy, but the scientific validity of this approach is questionable. While some data have shown an association between cardiac vagal tone and GI function, others have questioned this relationship [15,29,30]. Our study found that cardiac vagal function, as assessed by CARTs, was not associated with either GI transit times or the motility index. Despite the fact that the heart and the GI tract are regulated by sympathetic and parasympathetic nervous systems, there are differences in the length and physiology of the neurons responsible for their regulation [31]. This may lead to differences in how these nerves are affected. In addition, the sympathetic and parasympathetic nerves exert their effects on the GI tract indirectly through the enteric nervous system.

### 4.3. Clinical Implications of the Study

GI symptoms are common in people with T1D, but their correlation with measurable GI parameters is inconsistent [15]. Previous studies found that a longer colonic transit was correlated with constipation and postprandial fullness, while a decreased colonic motility index was correlated with diarrhea and decreased bloating in adults [15]. Our study also found that a low colonic motility index was associated with severe diarrhea and indigestion. The association between objective signs of gastroenteric neuropathy and time since the onset of T1D or suboptimal control of blood glucose suggests that early intervention is possible. As objective signs of GI neuropathy were not associated with other measures of autonomic neuropathy, patients at risk should be evaluated with specific GI tract methods. As a relatively simple, valid, and minor invasive method, WMC could serve that purpose. From adult patients with T1D, it is known that gastrointestinal dysmotility is common in the absence of GI symptoms [30].

Performing diagnostic tests on adolescents with T1D might be time-consuming and costly, and for that reason only recommended for adolescents with high-risk profiles such as long diabetes duration, poor metabolic control, abnormal screening tests, and/or GI symptoms (questionnaires). Annual screening may help with early detection. A widely recognized tool for monitoring bowel function is the Bristol Stool Chart (BSC), which has been shown to be associated with colonic transit in both healthy people and people with functional constipation [32]. However, to the best of our knowledge, it has not yet been shown in people with diabetes, who very often have extremely variable stool consistency within the same patient. Further research into screening methods and interventions is still needed. Early intervention will mainly be the optimization of glycemic controls, but detailed dietary recommendations could also be offered to some people with diabetes [7,33]. Likewise, improving gastric emptying time in patients with gastroparesis can potentially improve glycemic control [34].

Our findings on the high occurrence of GI symptoms among adolescents with T1D emphasize the need for addressing these symptoms in their treatment. Further research is necessary to explore the effectiveness of pharmacological interventions, such as prokinetics, laxatives, enzyme supplements, and antidiarrheal products, for managing motility dysfunction. Longitudinal studies evaluating the effect of tight glycemic control and lifestyle changes on GI symptoms are also needed. It has been reported that more pronounced acidity is present in people with T1D and peripheral neuropathy [35]. However, in our study, no differences in mean pH values in the different GI segments were found. The treatment of *Helicobacter pylori* gastritis has been attempted in children with T1D, but it was not found to improve metabolic control [36]. Therefore, this issue seemed to not require a special focus.

Gastroparesis increases glucose variability, especially during the night [37]. Thus, unexplained changes in glucose profiles should lead to investigations of GI function. Continuous glucose monitoring (CGM) has been suggested as a possible useful screening tool for detecting delayed gastric emptying [37], but further research is needed. Our results suggest that dysregulated adolescents with a reduced time in range and a longer diabetes duration seem to require additional attention. Further research of risk profiles for GI neuropathy, i.e., genetic, immunological, and microbiota profiles, should be performed [29]. Another subgroup that requires extra attention is adolescents with both T1D and celiac disease, as celiac disease increases risk for autonomic neuropathy by four times compared with the general population [38,39]. Our only participant with T1D and celiac disease supported this by having prolonged gastric emptying time.

### 4.4. Strengths and Limitations

The strength of our study is that it is the first to assess pan-intestinal function in adolescents with T1D. All tests were performed in a standardized manner by only two healthcare professionals, and our main findings were built on objective measurements rather than subjective reporting.

The limited population size and the higher proportions of female participants in the control group were limitations in our study. In addition, blood glucose levels were not checked during the test days, which could have affected the results, as hyperglycemia is known to slow gastric emptying [40]. However, the adolescents with T1D were informed to keep their blood glucose level in the target range, and they corrected their blood glucose during the test day if their CGM or insulin pump sounded an alarm, which reduced the impact of missing blood glucose measurements. Although HbA1c is not the correct factor to use, we did not find any association between HbA1c and gastric emptying. A limitation of WMC is that it lacks the capacity to detect abnormally rapid gastric emptying, which is a problem in the context of other methods, showing that rapid gastric emptying is frequently observed in adolescents with T1D [14], with important implications for postprandial glycaemic control. In addition, WMC gastric emptying times showed 52.8% agreement with scintigraphy, which is regarded as the gold-standard test [41].

In conclusion, objective signs of GI neuropathy are common in adolescents with T1D and are associated with the duration of disease and poor control of blood glucose. This calls for early intervention in patients at high risk of developing GI neuropathy.

## Figures and Tables

**Figure 1 jcm-12-01925-f001:**
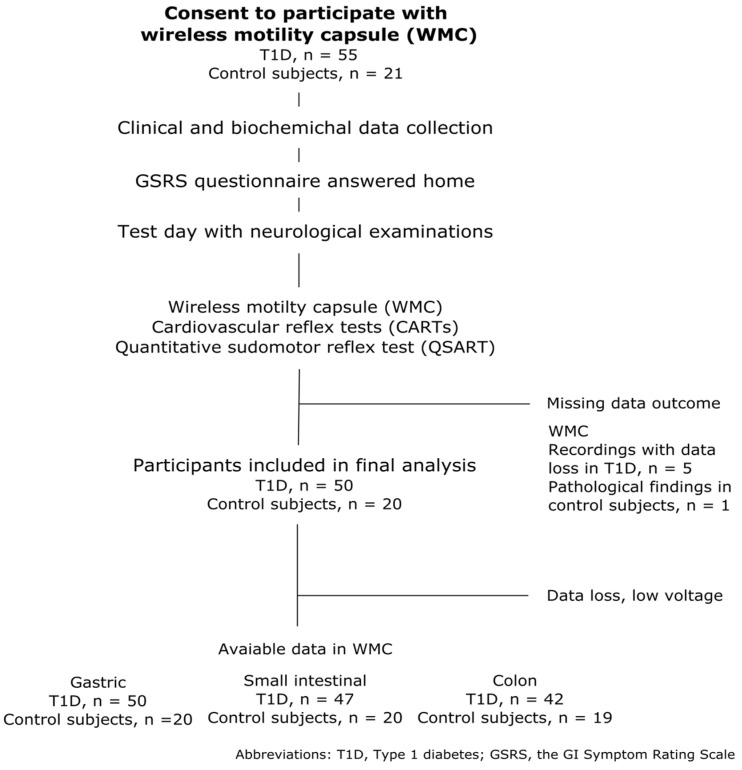
Flowchart of study population selection and the data available in the final analysis.

**Table 1 jcm-12-01925-t001:** Characteristics of the study population.

	Control, N = 20 ^1^	Diabetes, N = 50 ^1^	*p*-Value ^2^
Sex (female)	14 (70%)	24 (48%)	0.12
Age (Years)	16.60 (15.40–18.20)	16.95 (15.00–18.90)	0.26
Diabetes duration (Years)		8.5 (4.6–17.4)	
HbA1c (mmol/mol)	33 (27–40)	61 (41–93)	
BMI (kg/m^2^)	20.96 (17.95–30.40)	22.49 (17.63–29.61)	0.08
BMI-SDS	−0.1 (−1.0–1.7)	0.57 (−2.3–1.85)	<0.01
Height (cm)	173 (158–188)	174 (150–191)	0.65
Hip circumference (cm)	97 (65–112)	100 (82–114)	0.45
Waist circumference (cm)	74 (61–92)	75 (53–100)	0.63
Tanner (Stage)			
4	5 (25%)	13 (26%)	1.00
5	15 (75%)	37 (74%)	
SBP (mmHg)	112 (98–126)	116 (68–147)	0.18
DBP (mmHg)	70 (59–85)	77 (55–96)	<0.01
Pulse (beat per minute)	66 (55–90)	76 (50–106)	0.02
Cholesterol (mmol/L)	3.75 (2.80–5.10)	4.00 (3.00–6.40)	0.22
LDL (mmol/L)	2.05 (1.40–3.50)	2.10 (0.50–4.10)	1.00
HDL (mmol/L)	1.30 (0.68–2.20)	1.50 (0.97–3.70)	0.04
Triglycerides (mmol/L)	0.70 (0.30–1.10)	0.90 (0.30–3.80)	<0.01
Alcohol (units/week)			0.06
0	1 (5.0%)	5 (10%)	
1–3	17 (85%)	23 (46%)	
4–7	2 (10%)	14 (28%)	
8–14	0 (0%)	5 (10%)	
>15	0 (0%)	3 (6.0%)	
Smoking (Status)			0.81
Never	16 (80%)	38 (76%)	
Previous	3 (15%)	6 (12%)	
Smoke	1 (5.0%)	6 (12%)	
Activity (hours/week)			0.05
0	0 (0%)	5 (10%)	
1–3	2 (10%)	12 (24%)	
4–7	5 (25%)	18 (36%)	
>7	13 (65%)	15 (30%)	
Total daily insulin per weight per day (IE/kg/day))		0.86 (0.40–1.65)	
Basal insulin insulin per weight per day (IE/kg/day)		0.39 (0.14–0.87)	
Time in range (%)		52 (23–85)	
Time in hypoglycemia (%)		5.0 (0.0–15.0)	
Microvascular complication (%) ^†^			
retinopathy		2 (4%)	
nephropathy		2 (4%)	
Autoimmune disease (%)			
thyroid		5 (10%)	
celiac		1 (2%)	

^1^ Median (range) for continuous; n (%) for categorical. ^2^ Categorical variables, Fisher’s exact test; Continuous variable with normal-distribution, Welch two sample *t*-test; Continuous variable with non-normal-distribution, Wilcoxon rank sum test. Abbreviations: HbA1c, hemoglobin A1c; BMI, body mass index; SBP, systolic blood pressure; DBP, diastolic blood pressure; HDL, high-density lipoprotein; LDL low-density lipoprotein; NI, not indicated. ^†^ Retinopathy was based on an examination by an ophthalmologist, and nephropathy was defined as urine albumin to creatinine ratio of ≥30 mg/g.

**Table 2 jcm-12-01925-t002:** Comparison of gastrointestinal motility parameters and transit times obtained in adolescents with type 1 diabetes and healthy controls.

	**Controls ^1^**	**Diabetes ^1^**	** *p* ** **-Value ^2^**
**Motility Parameters**			
Gastric			
Pressure maximum (mmHg)	188 (25–408)	213 (45–379)	0.99
Mean peak amplitude (mmHg)	16.2 (12.3–22.0)	16.8 (13.0–23.4)	0.37
Contractions per minute (number)	1.7 (0.7–4.0)	1.5 (0.8–4.1)	0.79
Motility index (mmHg·second/min)	49 (17–109)	53 (26–354)	0.50
Small intestinal			
Pressure maximum (mmHg)	94 (50–187)	92 (37–372)	0.92
Mean peak amplitude (mmHg)	17.4 (13.8–22.9)	17.4 (13.8–22.8)	0.62
Contractions per minute (number)	5.3 (2.4–8.0)	5.0 (1.4–7.8)	0.49
Motility index (mmHg·second/min)	172 (74–389)	171 (41–338)	0.55
Colon			
Pressure maximum (mmHg)	116 (20–215)	138 (57–222)	0.16
Mean peak amplitude (mmHg)	17.7 (14.8–25.8)	19.7 (14.3–26.7)	0.01
Contractions per minute (number)	2.2 (1.5–4.7)	2.2 (0.6–6.0)	0.85
Motility index (mmHg·second/min)	131 (77–468)	195 (40–376)	0.02
**Transit times (min)**			
Gastric	230 (132–346)	195 (94–1177)	0.38
Antroduodenal transition	16 (1–57)	19 (1–106)	0.38
Small intestinal	334 (101)	296 (168–747)	0.47
Iliocecal transition	8 (1–65)	12 (1–65)	0.41
Colon	1446 (700–5813)	1283 (247–5456)	0.37
Whole gut	2089 (1296–6606)	1846 (610–6136)	0.36
**pH_median_**			
Gastric	2.0 (0.8–4.4)	1.5 (0.2–6.4)	0.11
Small intestinal	7.4 (6.4–7.9)	7.5 (4.8–7.9)	0.76
Colon	6.1 (5.4–7.5)	6.7 (4.0–7.8)	0.18

^1^ Data both with normal distribution and non-normal-distribution are presented as median (range). ^2^ Continuous variables with normal-distribution, Welch Two Sample *t*-test; Continuous variable with non-normal-distribution, Wilcoxon rank sum test.

## Data Availability

The datasets generated during and/or analyzed during the current study are not publicly available due to the General Data Protection Regulation but are available in an anonymized version from the corresponding author upon reasonable request.

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
