# Peer review of "Early Gastrointestinal Neuropathy Assessed by Wireless Motility Capsules in Adolescents with Type 1 Diabetes"

_jcm, 2023, doi:10.3390/jcm12051925_

Round 1
Reviewer 1 Report
To the authors; thank you for this nice and concise paper. Some minor editing is required.
My comments:
Line 55 (and also above) you underline the efficacy of targeted intervention, which could be a bit of a stretch. Could you perhaps be more specific, with some examples of this?
Line 73-74: Please check if the address for the producer is correct
Line 78 and in methods: It is unclear how you define “enteric neuropathy” and whether you use this as a synonym of GI autonomic neuropathy.
Line 140-142: Do you mean “giving” in these three lines?
Line 182: Do you mean compared to your controls? I.e. it might be more accurate to use that term in stead of normative data, which implies an existing database collected by certain principles.
Results: It is difficult to see the results for your 3rd aim: “3) to investigate possible risk factors for early GI autonomic 82 neuropathy in adolescents with T1D.” Could you perhaps summarize in 1-2 lines whether you found any relevant risk factors? Did you mention association with other microvascular complications in this group?
Results: Where are the results of the GSRS? If they were completely similar, this should be stated, also with median/mean values.
Line 229: exchange by with with
Discussion/results: how about the glucose levels at the time of examination: do you have data on that?
Line 295-297: Please check this sentence
Line 299: did you exclude celiac or investigate for unknown celiac? Please discuss / clarify
Reviewer 2 Report
The authors should be congratulated on accumulating a relatively large dataset in this group of patients who can be difficult to enrol in research studies.
The WMC has some appeal given its potential to evaluate motility in multiple gut segments, but its limitations are not adequately discussed. The capsule empties from the stomach generally with the return of phase III after a meal, and for the diagnosis of gastroparesis the agreement with scintigraphy (regarded as the gold standard) is only just over 50% (Hasler WL, Neurogastroenterol Motil. 2018;30(2): e13196). In particular, the WMC lacks the capacity to detect abnormally rapid gastric emptying, which is a problem in the context of other methods showing that rapid gastric emptying is frequently observed in adolescents with T1D (e.g. ref 13) – with important implications for postprandial glycaemic control.
Were all the study visits done in the morning? The Methods mention a 6 hour fast, but not specifically an overnight fast. This is a problem because there would be some variability in gut function from diurnal variation. I am also concerned that the Methods state that there was only a 2 hour fast from liquids. If nutrient-containing liquids were allowed up to 2 hours before testing, this would completely confound the outcomes.
I cannot see the outcomes of the autonomic function testing presented in the Results section. This seems an inexplicable omission.
The results in Table to give an SD in controls and a range in T1D patients. It would be helpful to present the distribution of data in a way that makes it easier to compare the 2 groups, i.e. is there much greater variability in T1D than in health?
No mention is made of the pancreatic polypeptide response to sham feeding as a marker of GI autonomic neuropathy.
It would also have been useful to have included the Bristol Stool Chart in the questionnaire, since there is evidence it has a good relationship with gut transit.
What kinds of “early intervention” do the authors envisage would eb prompted by these outcomes?
Round 2
Reviewer 2 Report
The manuscript has been improved by the revisions and y comments have been largely addressed. There are one or two minor issues of grammar and style that could be improved by proof reading by a fluent English writer.
Author Response
Dear Editor and Reviewers,
Thank you for accepting the revised version and our response to your initial comments. We are grateful for your suggestion to have a fluent English writer proofread it, which has now been done.
With these revisions, we hope the manuscript is now acceptable for publication.
On behalf of all authors
Vinni Faber Rasmussen, MD
Department of Clinical Medicine, Aarhus University, Denmark